# Derivation of Cell-Engineered Nanovesicles from Human Induced Pluripotent Stem Cells and Their Protective Effect on the Senescence of Dermal Fibroblasts

**DOI:** 10.3390/ijms21010343

**Published:** 2020-01-05

**Authors:** Hyelim Lee, Hyeonjin Cha, Ju Hyun Park

**Affiliations:** Department of Medical Biomaterials Engineering, Kangwon National University, 1 Kangwondaehak-gil, Chuncheon-si, Gangwon-do 24341, Korea; lath5070@naver.com (H.L.); hyeon_j03@kangwon.ac.kr (H.C.)

**Keywords:** cell-engineered nanovesicle (CENV), human induced pluripotent stem cell (iPSC), cellular senescence, skin aging

## Abstract

Stem cells secrete numerous paracrine factors, such as cytokines, growth factors, and extracellular vesicles. As a kind of extracellular vesicle (EV), exosomes produced in the endosomal compartment of eukaryotic cells have recently emerged as a biomedical material for regenerative medicine, because they contain many valuable contents that are derived from the host cells, and can stably deliver those contents to other recipient cells. Although we have previously demonstrated the beneficial effects of human induced potent stem cell-derived exosomes (iPSC-Exo) on the aging of skin fibroblasts, low production yield has remained an obstacle for clinical applications. In this study, we generated cell-engineered nanovesicles (CENVs) by serial extrusion of human iPSCs through membrane filters with diminishing pore sizes, and explored whether the iPSC-CENV ameliorates physiological alterations of human dermal fibroblasts (HDFs) that occur by natural senescence. The iPSC-CENV exhibited similar characteristics to the iPSC-Exo, while the production yield was drastically increased compared to that of iPSC-derived EVs, including exosomes. The proliferation and migration of both young and senescent HDFs were stimulated by the treatment with iPSC-CENVs. In addition, it was revealed that the iPSC-CNEV restored senescence-related alterations of gene expression. Treatment with iPSC-CENVs significantly reduced the activity of senescence-associated-β-galactosidase (SA-β-Gal) in senescent HDFs, as well as suppressing the elevated expression of p53 and p21, key factors involved in cell cycle arrest, apoptosis, and cellular senescence signaling pathways. Taken together, these results suggest that iPSC-CENV could provide an excellent alternative to iPSC-exo, and be exploited as a resource for the treatment of signs of skin aging.

## 1. Introduction

Skin aging is caused by intrinsic and extrinsic factors. Intrinsic aging is that involved in natural aging and age-dependence, while extrinsic aging is that caused by environmental factors, such as sunlight, and active oxygen [1]. The hallmarks of skin aging are wrinkles, roughness, and loss of elasticity, which are mainly due to changes in the extracellular matrix (ECM) materials, such as collagen, glycosaminoglycans, elastin, and hyaluronic acid with the progression of aging [2,3]. Fibroblast is the most common cell type in the lower compartment of skin connective tissue (dermis), and plays important roles in the maintenance of ECM and reconstitution of damaged skin tissues. Alterations in the biological functions of fibroblasts are responsible for the phenotypic changes in skin as a consequence of aging [4,5]. Several age-related alterations in fibroblasts have been reported: loss of protein homeostasis [6], down-regulation of ECM synthesis [7], and up-regulation of the expression of matrix-degrading enzymes, such as collagenase, hyaluronidase, and elastase [8]; decline in respiratory capacity and imbalanced reactive oxygen species (ROS) level by mitochondrial dysfunction [9], and DNA damage at chromosomal telomeres [10]. Cellular senescence is another hallmark of dermal fibroblasts in aging skin [11,12], and can be defined as irreversible cell cycle arrest activated via signaling pathways involving tumor suppressor p53, p21, and p16^INK4a^ [13,14,15]. Although cellular senescence plays a crucial role in preventing the propagation of genetically damaged and potentially oncogenic cells in young organisms, the increase in senescent cell population leads to the decline in cell replacement system for the regeneration of damaged tissues, because stem or progenitor cells start to lose their regenerative potential with aging [16,17,18]. 

Stem cells have the potential to differentiate into multiple types of cells (potency), and proliferate much more than proliferative somatic cells (self-renewal). Until now, stem cells and their secretomes have been widely used for the regeneration of damaged tissue in several clinical applications. Recently, several in vivo as well as in vitro studies have demonstrated that the treatment with exosomes derived from stem or progenitor cells ameliorates cellular functions and phenotypic symptoms in various injuries, including ischemia/reperfusion injury [19,20], myocardial infarction [21,22], and cutaneous closure [23,24,25]. The exosome released from multivesicular bodies (MVBs) with the plasma membrane is a small extracellular vesicle of 30–150 nm diameter that contains multiple host cell-derived molecules, such as proteins, peptides, messenger RNAs (mRNAs), micro RNAs (miRNAs), and long noncoding RNAs (lncRNAs) [26,27,28]. Most studies on the therapeutic application of stem cell-derived exosomes have focused on multipotent stem cells. However, in a previous study, we have demonstrated that human induced pluripotent stem cell-derived exosomes (iPSC-Exo) ameliorate genotypic and phenotypic changes of human dermal fibroblasts (HDFs) induced by photoaging and natural senescence [29]. Unlike adult stem and progenitor cells that have limited potential for self-renewal, pluripotent stem cells (PSCs), if cultured under well-established conditions, can be manipulated and maintain their physiological characteristics regardless of passage number, indicating that the controlled manufacture of exosomes is more suitable in PSCs than in adult stem cells [30,31]. However, the major drawback for further application of human PSC-derived exosomes has been believed to be the limited production yield of exosomes from donor cells. To obtain a sufficient amount of exosome from human PSCs, huge-scale culture is required, which is a very expensive and time-consuming process. 

To overcome the low production yield of iPSC-Exo, in this study we prepared cell-engineered nanovesicles (CENVs) from human iPSCs, and investigated their beneficial effects on naturally senescent HDFs. We utilized the serial extrusion method to produce the iPSC-CENVs, and purified them using density-gradient ultracentrifugation. The iPSC-CENV was verified to enclose human iPSC-specific mRNA, such as Oct4 and Nanog, and the production yield of iPSC-CENV was compared to that of extracellular vesicle (EV), including iPSC-Exo. Then, naturally senescent HDFs were treated with iPSC-CENVs to investigate whether iPSC-CENVs not only promote the proliferation and migration of cells, but also alleviate replicative senescence. The improvements of physiological alterations in senescent HDFs were evaluated using quantitative measurements of protein and genes involved in senescence. 

## 2. Results

### 2.1. Characterization of iPSC-CENVs

Figure 1 shows the experimental process for the preparation of CENVs from human iPSCs. iPSC-CENVs were generated by serial extrusion through polycarbonate membranes with diminishing pore sizes, and purified with density-gradient ultracentrifugation. 

The transmission electron microscope (TEM) images and the result for dynamic light scattering (DLS) analysis demonstrated that iPSC-CENVs had a spherical morphology (Figure 2A), and the average zeta potential was −15.8 mV (Figure 2B), respectively, suggesting that the isolated iPSC-CENV is an EV-like small nanoparticle, and enclosed in a negatively charged lipid bilayer consisting of human iPSC-derived plasma membrane. It has been reported that CENVs generated from certain cell types contain their specific markers [32,33]. To compare biological contents with cells and cell-derived nanovesicles, total RNAs were prepared from human iPSCs and iPSC-CENVs, respectively, and the RNA profiles were examined. Reverse transcriptase polymerase chain reaction (RT-PCR) analysis revealed that the iPSC-CENV contains pluripotent stem cell (PSC)-specific genes (Oct4, Nanog), as well as housekeeping gene (β-actin) (Figure 2C). 

### 2.2. Comparison of Productivity and Purity between iPSC-CENV and iPSC-EV Sample

The particle sizes and numbers were determined by using nanoparticle tracking analysis (NTA). We also isolated EVs secreted from human iPSCs from conditioned medium through ultracentrifugation method, and compared them with iPSC-CENVs. NTA results indicated that the diameter of iPSC-CENV was within the 100–250 nm range, and the modal value of the largest populations was 150.6 ± 4.0 nm (Figure 3A). Although the modal diameter was slightly higher and a small portion of large-size particles were included, iPSC-EVs exhibited a similar size distribution to iPSC-CENVs. This result is in line with the previous results of plasma membrane-derived nanovesicles prepared from other cell types, including several immortalized cell lines, primary cells, and stem cells [34,35,36,37]. Meanwhile, there was a little discrepancy in the size distribution of the iPSC-EVs compared to the iPSC-Exo used in our previous study [29], which is believed to be due to difference in isolation methods. We prepared iPSC-EVs by ultracentrifugation method in this study, while the iPSC-Exo was isolated using ExoQuick-TC^TM^ (Systems Bioscience, Palo Alto, CA, USA) in the previous study. Compared to the size distribution of iPSC-Exo, it was revealed that the iPSC-EVs contained more large-sized particles. However, the iPSC-EVs also contained a large proportion of particles with a size distribution that is similar to the iPSC-Exo, indicating that the iPSC-EVs mainly include exosomes. In addition, a recent study has demonstrated that human iPSC-derived EVs showed efficient internalization by recipient cells and alleviation of aging-associated phenotypes in senescent mesenchymal stem cells (MSCs) [38]. In this study, the EVs exhibited a size distribution that is similar to the iPSC-EVs of our present study rather than the iPSC-Exo. These results suggest that there is no significant difference in the beneficial effects of the iPSC-EVs on senescent cells compared to those of iPSC-Exo, and as a result, iPSC-EVs can be used as a control for comparison with iPSC-CENVs.

CENV provided comparative advantages over EV with regard to mass production. Comparing the total count of particles obtained from the same number of cells, the average yield of the iPSC-CENVs was 71.3 ± 2.0-fold higher than that of the EVs (Figure 3B). After the EVs were collected from the medium conditioned by human iPSCs for the last 24 h of culture, the iPSC-CENVs were generated from the remaining cells through the serial dilution method. If the iPSC-EVs were also collected from the conditioned medium at the early days of culture, the total production yield of EVs could be increased. However, not only was it a time-consuming and labor-intensive process, the increase would not have been considerable either, since there were few cells in the early stage of culture. Thus, it is apparent that even if the EVs were collected every day throughout the entire culture period, iPSC-CENVs would have been produced in significantly larger quantities compared to iPSC-EVs.

Then, in order to address the concern of cell-derived protein contaminants isolated along with iPSC-CENVs during the purification processes, the purities of both iPSC-CENV and iPSC-EV sample were assessed through the particle number-to-protein ratio, on the assumption that there was no significant difference in protein content between the single CENV and EV. The results revealed that iPSC-CENVs exhibited significantly higher purity than even the iPSC-EVs (more than 7-fold higher, Figure 3C). Taken together, these results suggest that not only do the iPSC-CENVs have similar physicochemical properties, such as size and surface charge, as the iPSC-EVs, the production yield is significantly improved, without any significant purity concern.

### 2.3. iPSC-CENV Promoted the Proliferation and Migration of HDFs

It was previously demonstrated that the iPSC-derived exosome accelerated the proliferation of HDFs [29]. To confirm the mitogenic effect of iPSCs-CENV, water-soluble tetrazolium salt-8 (WST-8) assay was performed to investigate the proliferation of HDFs (Figure 4A). Following serum starvation, young HDFs under passage number 5 were treated with different doses of iPSC-CENVs, ranging from particle numbers 0.25 to 4 × 10^9^/mL for 48 h. As expected, the proliferation of HDFs was significantly increased in a dose-dependent manner, similar to our previous result that revealed the effect of the iPSC-Exo. The proliferation was increased in the cells treated with particle number 2 × 10^9^/mL of iPSC-CENVs, by 172.2 ± 10.6% compared to the non-treated control cells, which was comparable with the previous result when the cells were treated with iPSC-Exo. In the same manner, we also investigated the effect of iPSC-CENVs on the proliferation of naturally senescent HDFs over passage number 28. Following 48 h treatment with particle number 2 × 10^9^/mL of iPSC-CENVs, the senescent HDFs revealed statistically significant increase in the proliferation, even though the increase was quite low compared to the result of young HDFs under passage number 5 (Figure 4B). 

We next examined the effect of iPSC-CENV on the migration of both young and senescent HDFs. In scratch closure assay, the time-course closure of scratched area demonstrated that the rate of gap-filling was significantly higher in the case of young cells than senescent cells. However, regardless of cellular senescence, treatment with iPSC-CENVs stimulated the migration of senescent HDFs, as well as young cells (Figure 5A). In a transwell migration assay, senescent HDFs were seeded on the upper side of a trans-membrane, and the medium in the lower chamber was replaced with serum-free DMEM/F12 containing particle number 2 × 10^9^/mL of iPSC-CENVs. The results of staining the cells that migrated to the opposite side across the porous membrane with crystal violet (Figure 5B), and quantification of the staining (Figure 5C), clearly indicated that iPSC-CENV stimulated the migration of senescent HDFs. Although the migratory potential of senescent cells was lower than that of young cells, as shown in the results of scratch closure assay, the migration-promoting effect of iPSC-CENV was comparable with the previous result of migration assays in which young HDFs were treated with the iPSC-Exo [29].

### 2.4. iPSC-CENV Reduced the Expression of Senescent-Related Genes

As cellular senescence progresses, somatic cells undergo diverse alterations in their phenotypic features, including flattened and enlarged morphology, limited capacity to proliferate, and several mitochondrial and metabolic abnormalities. In addition, many genetic alterations have been reported. SA-β-Gal is the most typical senescence marker, and exhibited age-dependent increase in dermal fibroblasts of skin biopsies provided from donors of different ages. Our previous study has demonstrated that the expression of SA-β-Gal was diminished by treatment with iPSC-Exo in senescent HDFs [29]. Here, we investigated whether iPSC-CENV could reduce SA-β-Gal activity in senescent HDFs. Figure 6A shows that senescent HDFs (over passage number 28, P28) expressed elevated SA-β-Gal activity, which can be visualized by staining at pH 6.0, compared to young cells (under passage number 5, P5). However, as with the previous results for iPSC-Exo, the histochemical SA-β-Gal activity was significantly reduced in the senescent HDFs by treatment with particle number 2 × 10^9^/mL iPSC-CENVs. For quantitative analysis, the number of clearly SA-β-Gal-positive cells in each experimental group was counted from three different microscopic images with total cells, and the percentages of stained cells were evaluated (Figure 6B). While a large population of SA-β-Gal-positive cells was observed in senescent HDFs at 62.4 ± 12.4%, this population was significantly reduced by treatment with iPSC-CENVs at 36.7 ± 9.8%. 

The tumor suppressor p53 is a potent transcription factor that can be regarded as a guardian of the genome, and contributes to suppress tumor formation by regulating diverse signaling networks involved in DNA repair, apoptosis, cell cycle arrest, and cellular senescence [39,40]. In addition, it has been reported that the increases in p53 expression, as well as its functional activity, induce cellular senescence, mainly via inducing p21 expression, an inhibitor of cyclin-dependent kinases (CDKs) that are responsible for cell cycle progression [41,42]. Numerous reports have demonstrated that p53/p21 pathway plays a pivotal role in cellular senescence [15,40,43,44]. Furthermore, altered expression of various types of collagen and matrix metalloproteinase (MMP) has been found to be associated with p53 expression in senescent cells [45,46]. Zhu et al. reported that knockdown of p53 in senescent cardiac fibroblasts of mouse heart increased the expression of collagen type I and III, but decreased the expression of MMP-2 and MMP-9 [47]. Thus, we next investigated whether CENV could suppress the p53/p21 signaling pathway. As a result of qPCR analysis, the mRNA expressions of both p21 and p53 were significantly reduced by treatment with particle number 2 × 10^9^/mL of iPSC-CENVs (Figure 7). Taken together, these results suggest that iPSC-CENV contributes to restoring cellular senescence by suppressing p53/p21 signaling pathway.

## 3. Discussion

Many studies have reported that exosomes derived from stem cells can be used for clinical applications, such as wound healing, regeneration of bone and heart tissues, etc. In the process of wound healing, scar is formed by tissues contraction. Fang et al. reported that human umbilical cord mesenchymal stem cell-derived exosomes (hUCMSC-Exo) inhibit scar formation by suppressing the transforming growth factor-β (TGF-β)/SMAD signaling pathway [48]. Tooi et al. studied the influence of human placenta mesenchymal stem cell-derived exosomes (hPlaMSC-Exo) on osteoblast differentiation by inducing the expression of Oct4 and Nanog, stemness-related genes [49]. In addition, exosomes play a role as carriers to deliver therapeutic drugs. Alvarez-Erviti et al. showed that siRNA-loaded exosomes could be delivered to brain, and reduced the expression of BACE1 gene, a target in Alzheimer’s disease [50]. Although exosome is a kind of resource for cell-based therapies, it is believed to overcome several obstacles caused by the direct clinical application of living cells, such as tumor formation, differentiation potential, handling, and transport conditions [51]. 

However, limited production yield, which is due to extremely low secretion level and extensive loss during the isolation process, has been considered an important drawback for the therapeutic application of stem cell-derived exosomes or EV. To overcome these obstacles, novel approaches to generate EV-mimetic nanovesicles by mechanically disrupting plasma membranes of various cell types have been introduced. Among them, the most representative method is the serial extrusion of cells through membrane filters with diminishing pore sizes. Jang et al. produced the CENVs in which chemotherapeutics were loaded from monocytes or macrophages using the serial extrusion method [37]. Kim et al. fabricated iron oxide nanoparticle-incorporated CENVs from human MSCs, and compared their therapeutic efficacy to that of normal CENVs for spinal cord injury treatment in a clinically relevant model [35]. In particular, some recent studies have reported the generation of pluripotent stem cell-derived CENVs and their biochemical effects. Park and colleagues generated CENVs from murine embryonic stem cell (mESC) line, and demonstrated that the mESC-derived CENV enhanced the proliferation rate of murine skin fibroblasts by stimulating the expression of genes and proteins involved in cell proliferation [52]. They also revealed that the mESC-derived CENV stimulates the signaling pathway related to proliferation in murine MSCs without any significant alterations in the multipotent characteristics of stem cells [33]. Previous studies confirmed that the artificial CENV holds similar biological contents, including membrane, cytoplasmic, nucleus proteins, and multiple types of RNAs to their donor cells. In addition, the therapeutic potential of the nanovesicles has been compared to that of exosomes or EVs derived from the same donor cells.

In the present study, we prepared CENVs from human induced pluripotent stem cells, and examined their effects on the proliferation, migration, and senescence-related gene expressions in naturally senescent HDFs. Our previous study demonstrated that iPSC-Exo not only promoted the proliferation and migration of young HDFs, but also exhibited protective effects on the damage and epigenetic alterations triggered by photoaging and natural senescence [29]. Compared with the iPSC-Exo, the treatment of iPSC-CENV promoted the proliferation and migration of HDFs in a similar manner (Figure 4 and Figure 5). In addition, iPSC-CENV showed significant stimulatory effects on cell proliferation and migration, even in senescent HDFs. Considered with the results indicating that iPSC-CENV down-regulated the expression of SA-β-Gal (Figure 6), a well-known senescence-associated marker and p53/p21, which are involved in cell cycle arrest and cellular senescence (Figure 7), it can be concluded that iPSC-CENV has therapeutic potential to ameliorate the replicative senescence of dermal fibroblasts. 

Apart from the biological effects of iPSC-CENV on cellular senescence, the most important advantage of iPSC-CENV over iPSC-Exo in further clinical applications is considered to be the high production yield. Compared with iPSC-EVs that mainly including exosomes, iPSC-CENVs indicated more than 70-fold higher production yield (Figure 3B). Meanwhile, preparation of CENVs had the concerns that a large amount of protein contaminants can be remained in the final sample even after the isolation process, which is due to the preparation process of CENVs involving cell disruption. In this study, iPSC-EVs were isolated using ultracentrifugation method. Since ultracentrifugation provides relatively high yield and purity of exosomes, it has been considered the most representative method for exosome isolation [53,54]. Some recent studies have revealed that with respect to nanovesicle purity, ultracentrifugation outperforms several commercialized isolation methods using precipitation solution [55,56]. Nevertheless, compared with iPSC-EVs, the final purity of the isolated iPSC-CENVs was significantly higher, indicating that the protein contamination in the isolated iPSC-CENVs sample was negligible, and might be an unnecessary anxiety (Figure 3C). 

Previous studies reported by Park and colleagues demonstrated that the RNA and protein contents of mESC-derived CENVs resemble those of the original mESCs [52,57]. The RNA profiles of mESC-derived CENVs showed no significant differences, compared to those of mESCs and exosomes, although small RNAs, such as tRNA and microRNA, were enriched in exosomes, compared to the others. In addition, the CENVs contained not only pluripotent-specific mRNAs with housekeeping gene, but also cytosolic, membrane, and nucleus proteins as do mESCs and exosomes, indicating that diverse cellular contents are successfully encapsulated into the CENVs, similar to exosomes. These previous studies, along with our present result showing that the human iPSC-derived CENVs contain iPSC-specific mRNAs, such as Oct4 and Nanog, consider that most cellular contents from human iPSCs are randomly encapsulated in the iPSC-CENVs, and the contents of iPSC-CENVs are comparable with those of iPSC-Exo, despite the different quantities of each component. 

In conclusion, our present study has demonstrated that the human iPSC-derived CENVs exhibited similar physicochemical features to iPSC-Exo, and ameliorated physiological deteriorations and subsequent epigenetic alterations that occurred by natural senescence. Although the effect of murine ESC-derived CENV on the proliferation of primary murine skin fibroblasts has been reported, to the best of our knowledge, this study is the first to examine the effect of human pluripotent stem cell-derived CENV on the natural senescence of human skin fibroblasts. High production yield is an important advantage of iPSC-CENV compared to iPSC-Exo for clinical applications, even though the beneficial effects for skin aging are expected to be similar. These results suggest that the iPSC-CENV could provide a technical advance, and be used as a versatile tool in regenerative medicine for the treatment of skin aging. 

## 4. Materials and Methods

### 4.1. Cell Culture

As described in our previous study, human iPSC line was provided by the National Stem Cell Bank of Korea (Korea National Institute of Health) [29]. This cell line was generated by transducing Oct4, Sox2, cMyc, and Klf4 genes (Yamanaka factors) into HDFs using Sendai Virus. The cells were cultured using TeSR–E8™ medium (StemCell Technologies, Vancouver, BC, Canada), and the medium was replaced daily. When the confluency reached 70–80%, human iPSCs were detached by treatment with 0.5 mM EDTA, and re-plated onto a Matrigel™ (Corning, Corning, NY, USA)-coated dish. HDFs were maintained in Dulbecco’s modified Eagle’s medium (DMEM, Hyclone, Logan, UT, USA) supplemented with 10% fetal bovine serum (FBS, Welgene, Daejeon, Korea), 100 U/mL penicillin, and 100 µg/mL streptomycin (growth medium). All cultures were performed in a humidified incubator with 5% CO_2_ at 37 °C.

### 4.2. Preparation of iPSC-Derived EV and CENV

iPSC-CENVs were generated from human iPSCs according to the serial extrusion method established in previous studies [42]. The iPSCs were suspended in phosphate buffered saline (PBS), and serially extruded nine times through 10, 5, and 1 µm polycarbonate track-etched membrane filters (GVS, Bologna, Italy), respectively, using a mini-extruder (Avanti Polar Lipids, Birmingham, AL, USA). For removal of cell debris, the suspension was centrifuged at 500× *g* for 5 min, and 10,000× *g* for 30 min, in a stepwise manner. For density-gradient ultracentrifugation, 1 mL of 10% iodixanol (OptiPrep™, Axis-Shield, Oslo, Norway) was carefully layered on top of 1 mL of 50% iodixanol in a 5 mL ultracentrifuge tube (Beckman Coulter, Fullerton, CA, USA). The precleaned supernatants were overlaid on top of the 10% iodixanol, and centrifuged at 100,000× *g* for 2 h at 4 °C. The colored layer was collected from the interface of 50% and 10% iodixanol, and subsequently centrifuged again at 100,000× *g* for 2 h at 4 °C. Then, the iPSC-CENVs obtained from the pellet fraction were resuspended in PBS. To prepare EVs derived from the iPSCs, conditioned media were collected at the last day of subculture, and centrifuged at 500× *g* for 5 min to remove cellular debris. Following centrifugation at 100,000× *g* for 2 h, the pellet that contains EVs was resuspended in PBS after washing once. Finally, both iPSC-CENVs and iPSC-EVs were stored at −70 °C, until use. 

### 4.3. Characterization of iPSC-CENV

The concentration and size distribution of iPSC-CENVs were measured by nanoparticles tracking analysis (NTA), using NanoSight NS300 (Malvern Panalytical, Malvern, UK). For NTA, the iPSC-CENVs were diluted in PBS, and injected into the laser chamber using 1 mL syringe. The mode and concentration of iPSC-CENVs were determined by NTA 3.0 software. Each experiment was carried out in triplicate. Zeta potential was measured using Zetasizer Nano ZS (Malvern Panalytical). The morphologies of iPSC-CENV were visualized by transmission electron microscopy (TEM) using JEM-2100F (JEOL, Tokyo, Japan). The iPSC-CENVs were dried on a formvar/ carbon-coated grid, and negatively stained with 2% uranyl acetate for 10 min. Samples were observed by TEM operated at 60 kV.

### 4.4. RNA Isolation and Reverse Transcriptase PCR (RT-PCR)

Total RNA was isolated using a Ribospin™ total RNA purification kit (GeneAll Biotechnology Co, Ltd., Seoul, Korea), as described in previous study [29]. Briefly, DNA was synthesized from purified total RNA using TOPscript RT DryMIX (Enzynomics Co, Ltd., Daejeon, Korea) with dT18 plus primer. For RT-PCR, the amplification of the cDNA sample was performed using the PerfectShot™ Ex Taq (Loading dye mix, Takara, Shiga, Japan) with 30 cycles of denaturation at 94 °C for 30 s, annealing at 56 °C (for Oct4, Nanog) or 58 °C (β-actin) for 30 s, and elongation at 72 °C for 30 s. Quantitative real-time RT-PCR (qPCR) was also carried out using the Eco Real-Time PCR System (Illumina, San Diego, CA, USA) with TOPreal™ qPCR 2× PreMIX (SYBR Green with high ROX, Enzynomics Co. Ltd., Daejeon, Korea). Table 1 summarizes the sequences of primers used in RT-PCR analysis. The expression level of specific mRNA was normalized by that of β-actin as an endogenous control.

### 4.5. WST-8 Assay

Cell proliferation was determined by water-soluble tetrazolium salt-8 (WST-8) assay using Quanti-MAX™ WST-8 cell viability assay kit (Biomax Inc., Seoul, Korea), according to the manufacturer’s instructions. HDFs were plated at 2 × 10^4^ cells/cm^2^ in 96-well plate, and incubated in growth medium for 24 h. After a further 24 h incubation in DMEM containing 0.5% FBS for serum starvation, cells were treated with particle numbers 0.25, 0.5, 1, 2, and 4 × 10^9^/mL of iPSC-CENVs in serum-free DMEM/F12 for 48 h, respectively. Subsequently, 10 µL of WST-8 solution was added to each well, and incubated for 2 h. Absorbance at 450 nm was measured by microplate reader.

### 4.6. Cell Migration Assay

For scratch closure assay, HDFs were plated at 4 × 10^4^ cells/cm^2^ in 24-well plate, and incubated in growth medium for 24 h. After serum starvation for 24 h, the monolayer of cells was scratched with a sterile pipette tip. Subsequently, cells were incubated with iPSC-CENVs in serum-free DMEM/F12 for the indicated time, and closure of the scratched area by cell migration was monitored under an optical microscope. For transwell migration assay, HDFs were plated onto upper chamber of transwell (Corning, Lowell, MA, USA), and incubated in growth medium for 24 h. Following the treatment with iPSC-CENVs for 48 h in serum-free DMEM/F12, both cells that remained on the upper side and those that had migrated toward the opposite side of the trans-membrane were fixed by 4% paraformaldehyde. After removal of the cells on the upper side by cotton swap, the cells at the bottom side of membrane were stained with 0.5% crystal violet (Sigma-Aldrich, St. Louis, MO, USA) for 15 min, and observed under an optical microscope. For quantitative analysis, the membrane was treated with 50% acetic acid to dissolve the crystal violet, and the optical absorbance at 560 nm was measured using a microplate reader.

### 4.7. SA-β-Gal Staining

SA-β-Gal staining was conducted using senescent cells histochemical staining kit (Sigma-Aldrich). HDFs were plated in 96-well plate, and incubated in growth medium for 24 h. Following treatment with the indicated concentration of iPSC-CENVs for 48 h, cells were further incubated in serum-free DMEM/F12 for another 48 h. After washing with PBS, cells were fixed with 4% paraformaldehyde, and SA-β-Gal was stained by treatment with 100 µL staining mixture. Three images of different sites per each well plate were collected, and SA-β-Gal-stained cells were counted. The percentage of senescent cells were determined by dividing the number of stained cells by the total number of cells.

### 4.8. Statistical Analysis

All quantitative results were expressed as mean ± standard deviation. Statistical significance was determined by Student’s *t*-test. A value of *p* < 0.05 was considered significant.

## Figures and Tables

**Figure 1 ijms-21-00343-f001:**
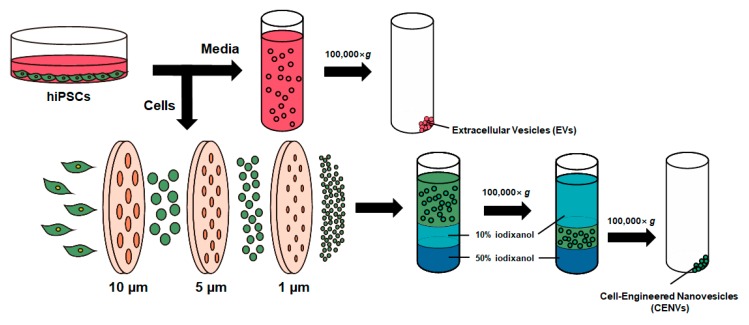
Schematic of the preparation of extracellular vesicles (EVs), and cell-engineered nanovesicles (CENVs) from human induced pluripotent stem cells (iPSCs). For iPSC-EVs preparation, spent medium conditioned by human iPSCs was collected, and subsequently ultracentrifuged for 2 h. For iPSC-CENVs preparation, cells were serially extruded through the polycarbonate membrane with diminishing pore size. Then, the generated iPSC-CENVs were collected by density-gradient ultracentrifugation using iodixanol (OptiPrep™).

**Figure 2 ijms-21-00343-f002:**
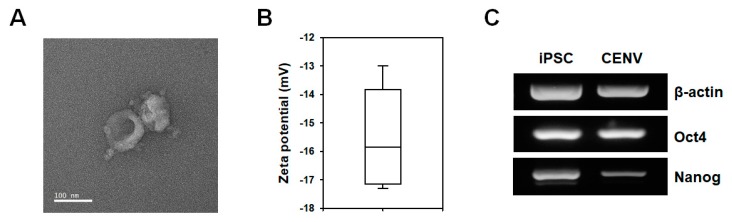
Characterization of iPSC-CENV. (**A**) Transmission electron microscopy (TEM) image of iPSC-CNEV (scale bar = 100 nm). (**B**) Dynamic light scattering (DLS) data of iPSC-CENVs. The mean value of zeta potential was −15.8 mV. (**C**) Reverse transcriptase polymerase chain reaction (RT-PCR) analysis of iPSCs and iPSC-CENVs for housekeeping gene (β-actin) and pluripotency-specific genes (Oct4, Nanog).

**Figure 3 ijms-21-00343-f003:**
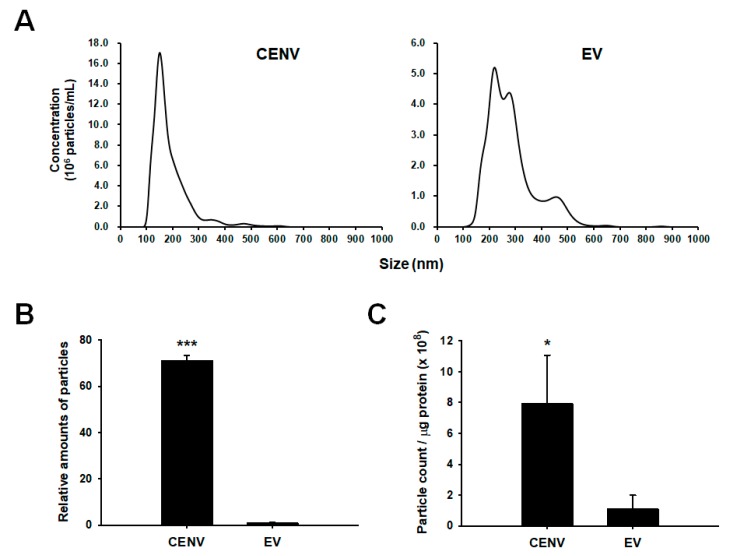
Comparison of human iPSC-derived CENV and EV. (**A**) Size distribution of iPSC-CENVs and iPSC-EVs, as evaluated by nanoparticle tracking analysis (NTA). (**B**) The number of particles isolated from the same number of human iPSCs, as evaluated by NTA. The number of iPSC-CENVs was normalized by that of iPSC-EVs, and represented as a relative value. (**C**) The particle-to-protein ratio of iPSC-CENVs and iPSC-EVs, indicating the relative purity against protein contaminants. Statistical significance (* *p* < 0.05, *** *p* < 0.005 compared to EVs) was determined using Student’s *t*-test. Error bars indicate the standard deviation of triplicate experiments.

**Figure 4 ijms-21-00343-f004:**
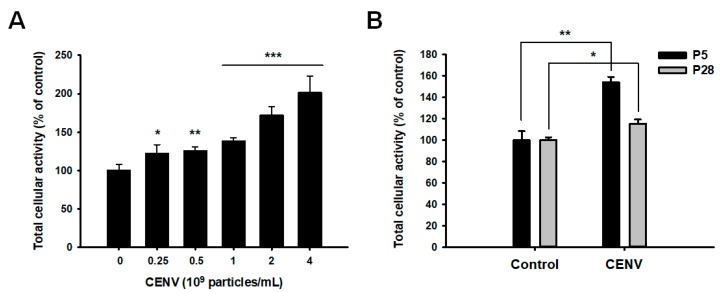
Effect of iPSC-CENV on the proliferation of human dermal fibroblasts (HDFs). (**A**) After serum starvation, cells were treated with iPSC-CENVs for 48 h in serum-free condition. The population of active HDFs was determined by water-soluble tetrazolium salt-8 (WST-8) assay. The value of iPSC-CENV-untreated cells was regarded as 100%, and the populations of other treated groups were represented as relative values. (**B**) Serum-starved young (under passage number 5, P5) and senescent (over passage number 28, P28) HDFs were treated with particle number 2 × 10^9^/mL of iPSC-CENVs for 48 h in serum-free condition, respectively, then the active cell populations were determined by WST-8 assay. Statistical significance (* *p* < 0.05, ** *p* < 0.01, and *** *p* < 0.005 compared to the iPSC-CENV-untreated group) was determined using Student’s *t*-test. Error bars indicate the standard deviation of triplicate experiments.

**Figure 5 ijms-21-00343-f005:**
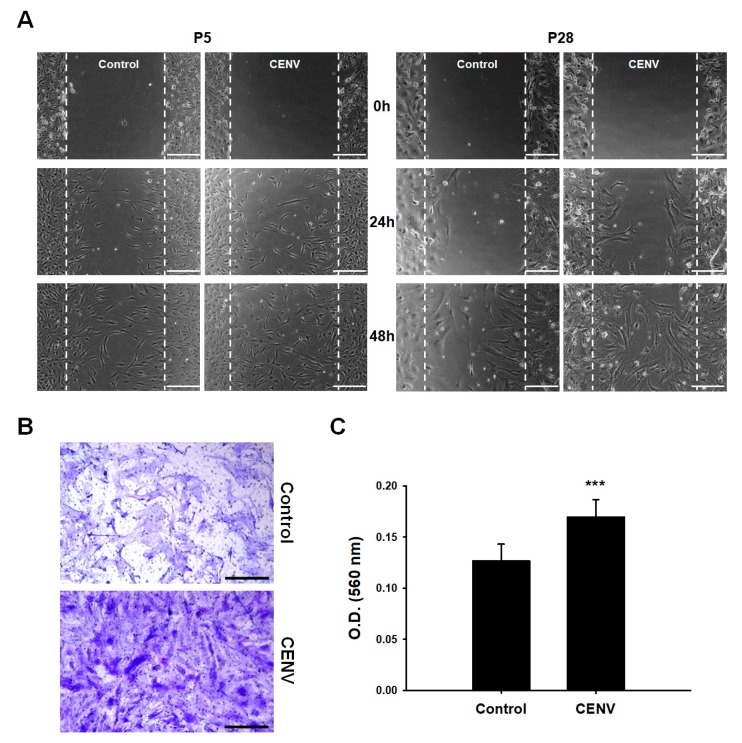
Effect of iPSC-CENV on the migration of HDFs. (**A**) Scratch closure assay for HDFs treated with iPSC-CENVs (scale bar = 250 µm). Serum-starved HDFs were treated with particle number 2 × 10^9^/mL of iPSC-CENVs. Dashed lines indicate the initial scratched area. The rate of gap-filling was estimated using phase-contrast microscopy. (**B**) Representative image of a transwell migration assay (scale bar = 250 µm). Senescent HDFs seeded onto the upper side of a porous membrane migrated to the bottom side through the membrane. After 24 h treatment with iPSC-CENVs, the cells that had passed through a porous membrane were fixed with 4% paraformaldehyde, and stained with crystal violet. (**C**) Quantitative analysis of a transwell migration assay. After removing the remaining cells on the upper side, the porous membrane with the stained cells was cut out, and immersed in a 50% acetic acid solution to dissolve the crystal violet. The intensity of staining was measured using spectrophotometry as absorbance at 560 nm. Statistical significance (*** *p* < 0.005 compared to the iPSC-CENV-untreated group) was determined using Student’s *t*-test. Error bars indicate the standard deviation of triplicate experiments.

**Figure 6 ijms-21-00343-f006:**
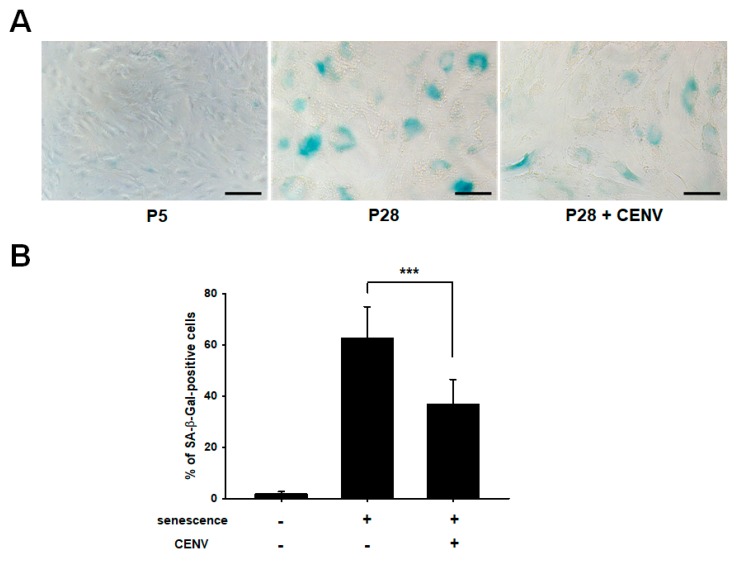
iPSC-CENV reduced the expression of senescent associated β-galactosidase (SA-β-Gal). (**A**) SA-β-Gal-positive cells were stained using senescent cells histochemical staining kit, and observed under phase-contrast microscopy (scale bar = 100 µm). (**B**) Quantitative analysis of SA-β-Gal positive cells. Each number of total cells and SA-β-Gal-stained cells was counted from three different microscopic images per each well plate, and the percentage of SA-β-Gal-positive cells was represented. Statistical significance (*** *p* < 0.005 compared to the iPSC-CENV-untreated group) was determined using Student’s *t*-test. Error bars indicate standard deviation of three independent well plates in a single representative experiment.

**Figure 7 ijms-21-00343-f007:**
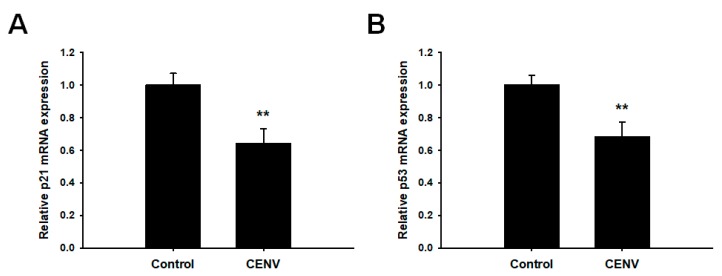
Quantitative analysis of (**A**) p21, and (**B**) p53, expression in senescent HDFs. The mRNA expression levels for each gene were analyzed by quantitative real-time PCR. Each mRNA sample was obtained from senescent HDFs treated with particle number 2 × 10^9^/mL of iPSC-CENVs for 48 h and non-treated cells, respectively. Statistical significance (** *p* < 0.01 compared to the non-treated control) was determined using Student’s *t*-test. Error bars indicate the standard deviation of triplicate experiments.

**Table 1 ijms-21-00343-t001:** List of primers for RT-PCR.

Gene	Primer	Sequence (5′-3′)
β-actin	Sense	GTG GGG CGC CCC AGG CAC CA
	Antisense	CTC CTT AAT GTC ACG CAC GAT TT
Oct4	Sense	TGT ACT CCT CGG TCC CTT TC
	Antisense	TCC AGG TTT TCT TTC CTA GC
Nanog	Sense	CAA AGG CAA ACA ACC CAC TT
	Antisense	ATT GTT CCA GGT CTG GTT GC
p53	Sense	GCC CAA CAA CAC CAG CTC CT
	Antisense	CCT GGG CAT CCT TGA GTT CC
p21	Sense	GAC ACC ACT GGA GGG TGA CT
	Antisense	CAG GTC CAC ATG GTC TTC CT

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
