# Peer review of "Derivation of Cell-Engineered Nanovesicles from Human Induced Pluripotent Stem Cells and Their Protective Effect on the Senescence of Dermal Fibroblasts"

_ijms, 2020, doi:10.3390/ijms21010343_

Round 1

Reviewer 1 Report

The manuscript entitled "Derivation of Cell-Engineered Nanovesicles from Human Induced Pluripotent Stem Cells and Their Protective Effect on the Senescence of Dermal Fibroblasts”, by Lee et al. describes the beneficial effects of cell-engineered nanovesicles (CENVs), generated by serial extrusion of human iPSCs through membrane filters with diminishing pore sizes, on young and senescent human dermal fibroblasts (HDFs). These nanovesicles, which can be produced with a relatively high yield, may be another promising alternative therapy to regenerate some tissues or deliver therapeutic drugs.

The paper fits well in the scope of IJMS, and complete the previous publication of the same team, in IJMS last year where they demonstrated similar beneficial effect on the aging of skin fibroblasts after their treatment with human induced potent stem cell-derived exosomes (iPSC-Exo).

However, the paper suffers from several flaws, which need to be carefully addressed before the paper can be considered for publication.

General comments:

Lane 103-105: Authors wrote: “Reverse transcriptase polymerase chain reaction (RT-PCR) analysis revealed that the iPSC-CENV contains pluripotent stem cell (PSC)-specific genes (Oct4, Nanog), as well as housekeeping gene (β-actin), in a similar manner to the iPSC”.

“In a similar manner to the iPSC” is confusing. Is it means they have the same level of mRNA? Because it is only qualitative. Authors should check the RNA level by quantitative RT-PCR.

It also be good to add negative controls (DNA, ...).

RNAseq could be a good way to compare mRNA from iPSC and the one of CENV.

As authors compare iPSC-CENVs to iPS-EV, could authors add mRNA expression of iPS-EV?

Moreover, in a therapeutic point of view, it will be interesting to check the expression of Oct4 and Nanog in HDFs treated with iPSC-CENVs (and untreated HDFs in control). Is Oct4 and Nanog mRNA from iPSC-CENVs enter in the HDFs.

Lane 138, Figure 3A: The size distribution of iPSC-CENVs and iPSC-EVs have been evaluated by nanoparticle tracking analysis (NTA). Could authors clarify why the size distribution of EV are largely different from the size distribution of iPSC-EVs that authors previously showed in Figure 1A of the study published in 2018 in IJMS (“Exosomes Derived from Human Induced Pluripotent Stem Cells Ameliorate the Aging of Skin Fibroblasts”). If the preparation is not similar between iPSC-EVs and iPSC-Exo, it seems inappropriate to compare results of this study to the ones of the previous one.

Line 185, Figure 5B: Staining is difficult to interpret because of the strong signal (background), especially in the control panel. Nuclei were better stained on the pictures of the previous publication. Please replace these pictures.

Lane 212 and 220, Figure 6: Number of independent experiment is not clear. Indeed, it is not clear if the 3 different images are from three distinct experiments or from different wells of the same experiment. If cells have been counted from three different microscopic images of a same experiment, it is not a triplicate experiments and it is necessary to reproduce independent experiments.

In addition, is it possible to precise the total number of counted cells? High number of counted cells would provide a better quantitative analysis.

Moreover, in order to better see the proportion of stained cells, pictures should be done at a lesser magnification.

Lane 283 and Figure 7: Authors demonstrated that treatment of senescent HDF decrease the expression of P53/p21 which are involved in cell cycle arrest and cellular senescence. Does expression of Ki67 or other proliferative markers have been investigated? Is Ki67 is increased by iPSC-CENV?

Moreover, it would be informative to have the expression of p21 and p53 in young HDFs. Indeed, young HDFs were taken as control in all figures except this one.

Minor comments:
1. Lane 118 to 120: “This result is in line with the previous results of plasma membrane-derived nanovesicles prepared from other cell types, including several immortalized cell lines, primary cells, and stem cells”: add references.

Figure 3A: It would be more readable if the scale of the Y axis would be similar between both graphs.

Moreover, authors may compare the area under the curve.

Author Response

General comments:

Lane 103-105: Authors wrote: “Reverse transcriptase polymerase chain reaction (RT-PCR) analysis revealed that the iPSC-CENV contains pluripotent stem cell (PSC)-specific genes (Oct4, Nanog), as well as housekeeping gene (β-actin), in a similar manner to the iPSC”.

“In a similar manner to the iPSC” is confusing. Is it means they have the same level of mRNA? Because it is only qualitative. Authors should check the RNA level by quantitative RT-PCR.

It also be good to add negative controls (DNA, ...)

RNA-seq could be a good way to compare mRNA from iPSC and the one of CENV.

As authors compare iPSC-CENVs to iPS-EV, could authors add mRNA expression of iPS-EV?

Moreover, in a therapeutic point of view, it will be interesting to check the expression of Oct4 and Nanog in HDFs treated with iPSC-CENVs (and untreated HDFs in control). Is Oct4 and Nanog mRNA from iPSC-CENVs enter in the HDFs.

<Answer>

First of all, we agree with the reviewer’s comment that “in a similar manner to the iPSC” is confusing. As the reviewer pointed out, we qualitatively compared the mRNAs of iPSCs and iPSC-CENVs in the present study. When cells were extruded through micro-sized pores, the plasma membrane was disrupted by adhesive tension, and the cellular contents were released into the aqueous phase. As fragments of lipid bilayer form into a spherical vesicle (CENV) by self-assembly in the aqueous phase, some fraction of the cellular contents can be encapsulated in the CENV, while others would be remained in the aqueous phase. Yoon et al. have quantitatively evaluated the amount of total RNAs from murine ESC-derived NVs and compared with that from murine embryonic stem cells (mESCs) used in the NV generation [1]. They demonstrated that the total RNAs in NVs generated from 1 × 106 mESCs were less than 1/10 of the RNAs isolated from the same number of cells, although the NVs were generated by a novel cell-slicing system rather than serial exclusion method in this study. For the reason, the amount of RNAs is supposed to be lower in iPSC-CENVs than that from an equivalent number of iPSCs. In our present study, however, it was clearly demonstrated that the iPSC-CENVs encompass the iPSC-specific cellular contents such as Oct4 and Nanog mRNA, and ameliorate the natural senescence of HDFs. Along with high production yield, these results are considered the main findings of this study.

As described in Introduction and Discussion sections, several studies have reported that CENVs generated from various types of cells by serial exclusion method contain host cell-derived cellular contents such as RNAs and proteins. Although we have not found any previous reports examining the DNA content of CENVs, it is considered that no significant levels of DNA release occur during cell membrane disruption because the chromosomal DNA of cell nucleus is tightly packed with chromatin structures. Global profiling of RNA expression and genomic DNA analysis using next generation sequencing tools could provide a technical advance toward further clinical applications of iPSC-CENVs. We have a plan to conduct those quantitative assessments in further in vivo study using the iPSC-CENVs.

Meanwhile, numerous studies have demonstrated that exosomes, EVs, or CENVs enclosed in a lipid bilayer can easily penetrate into recipient cells and deliver their cellular contents as described in the manuscript. Accordingly, it is believed that iPSC-specific mRNAs of iPSC-CENVs such as Oct4 and Nanog can be delivered to HDFs.

Yoon, J.; Jo, W.; Jeong, D.; Kim, J.; Jeong, H.; Park, J., Generation of nanovesicles with sliced cellular membrane fragments for exogenous material delivery. Biomaterials 2015, 59, 12-20.

In accordance with the reviewer’s comment, we carefully revised the sentence pointed out by the reviewer as follow;

(Page 3, Line 104)

“Reverse transcriptase polymerase chain reaction (RT-PCR) analysis revealed that the iPSC-CENV contains pluripotent stem cell (PSC)-specific genes (Oct4, Nanog), as well as housekeeping gene (β-actin) (Figure 2C).”

Lane 138, Figure 3A: The size distribution of iPSC-CENVs and iPSC-EVs have been evaluated by nanoparticle tracking analysis (NTA). Could authors clarify why the size distribution of EV are largely different from the size distribution of iPSC-EVs that authors previously showed in Figure 1A of the study published in 2018 in IJMS (“Exosomes Derived from Human Induced Pluripotent Stem Cells Ameliorate the Aging of Skin Fibroblasts”). If the preparation is not similar between iPSC-EVs and iPSC-Exo, it seems inappropriate to compare results of this study to the ones of the previous one.

<Answer>

As the reviewer pointed out, we prepared iPSC-EVs by ultracentrifugation method in the present study, while iPSC-Exo was isolated using ExoQuick-TCTM in our previous study (Oh et al., 2018, IJMS). Compared to the size distribution of iPSC-Exo in the previous study, it was revealed that iPSC-EVs contained more large-size particles. However, iPSC-EVs also contained a large proportion of particles with a size distribution similar to iPSC-Exo. In addition, Liu et al. reported the alleviative effect of human iPSC-derived EVs on cellular senescence in a recent study [2]. In this study, the EVs were purified using centrifugal filters and size exclusion chromatography, and exhibited a size distribution similar to that of iPSC-EVs isolated in our present study, rather than iPSC-Exo of the previous study. They demonstrated that the human iPSC-derived EVs showed efficient internalization by recipient cells and alleviation of aging-related phenotypic changes such as delayed cell growth, and elevated SA-β-Gal activity and p53/p21 gene expressions in senescent mesenchymal stem cells (MSCs). Although there are some differences in the senescent cell types (HDFs and MSCs) and analytical methods, both human iPSC-derived EVs in their study and iPSC-Exo in our previous study have been shown to similarly ameliorate aging of human cells. According to these results, we consider that there are no significant differences in the beneficial effects of iPSC-EVs on senescent HDFs compared to those of iPSC-Exo, and as a result, the iPSC-EVs can be used as a control for comparison with iPSC-CENVs.

Liu, S.; Mahairaki, V.; Bai, H.; Ding, Z.; Li, J.; Witwer, K.W.; Cheng, L., Highly purified human extracellular vesicles produced by stem cells alleviate aging cellular phenotypes of senescent human cells. Stem Cells 2019, 37, 779-790.

In accordance with the reviewer’s comment, we added the following sentences in Results section and carefully revised the paragraph;

(Page 4, Line 121)

“Meanwhile, there was a little discrepancy in the size distribution of the iPSC-EVs compared to the iPSC-Exo used in our previous study [29], which is believed to be due to difference in isolation methods. We prepared iPSC-EVs by ultracentrifugation method in this study, while the iPSC-Exo was isolated using ExoQuick-TCTM (System Biosciences, Palo Alto, CA, USA) in the previous study. Compared to the size distribution of iPSC-Exo, it was revealed that the iPSC-EVs contained more large-size particles. However, the iPSC-EVs also contained a large proportion of particles with a size distribution similar to the iPSC-Exo, indicating that the iPSC-EVs mainly include exosomes. In addition, a recent study has demonstrated that human iPSC-derived EVs showed efficient internalization by recipient cells and alleviation of aging-associated phenotypes in senescent mesenchymal stem cells (MSCs) [38]. In this study, the EVs exhibited a size distribution similar to the iPSC-EVs of our present study rather than the iPSC-Exo. These results suggest that there is no significant difference in the beneficial effects of iPSC-EVs on senescent cells compared to those of iPSC-Exo, and as a result, iPSC-EVs can be used as a control for comparison with iPSC-CENVs.”

Line 185, Figure 5B: Staining is difficult to interpret because of the strong signal (background), especially in the control panel. Nuclei were better stained on the pictures of the previous publication. Please replace these pictures.

<Answer>

  We agree the reviewer’s opinion. In accordance with the reviewer’s comment, we replaced the pictures of crystal violet-staining (Figure 5B).

Lane 212 and 220, Figure 6: Number of independent experiments is not clear. Indeed, it is not clear if the 3 different images are from three distinct experiments or from different wells of the same experiment. If cells have been counted from three different microscopic images of a same experiment, it is not triplicate experiments and it is necessary to reproduce independent experiments.

In addition, is it possible to precise the total number of counted cells? High number of counted cells would provide a better quantitative analysis.

Moreover, in order to better see the proportion of stained cells, pictures should be done at a lesser magnification.

<Answer>

For a single SA-β-Gal staining experiment, HDFs were plated in three distinct well plates per each experimental group and took three images of different sites per each well. Then, we counted the number of all the cells in these three images and regarded this number as the experimental value for a single well. Across these three different images, we counted more than 450 cells per each well as a result. For statistical analysis, we repeated these process for three distinct well plates, obtaining three independent data sets, and using them to evaluate the mean value and standard deviation for each experimental group (young, senescent without iPSC-CENV, and senescent with iPSC-CENV). We have performed SA-β-Gal staining experiments several times on different dates and observed positive effect of iPSC-CENVs each times. Figure 6 shows one representative result of those experiments.

Meanwhile, we agree the reviewer’s suggestion on the magnification of the pictures. However, when cells were observed at higher magnification, it was easier to recognize SA-β-Gal-positive cells.

In accordance with the reviewer’s comment, we corrected the sentence in Figure Legend as follows;

(Page 8, 233)

“Each number of total cells and SA-β-Gal-stained cells was counted from three different microscopic images per each well plate, and the percentage of SA-β-Gal-positive cells was represented.”

(Page 8, Line 236)

“Error bars indicate standard deviations of three independent well plates in a single representative experiment.”

Lane 283 and Figure 7: Authors demonstrated that treatment of senescent HDF decrease the expression of P53/p21 which are involved in cell cycle arrest and cellular senescence. Does expression of Ki67 or other proliferative markers have been investigated? Is Ki67 is increased by iPSC-CENV?

Moreover, it would be informative to have the expression of p21 and p53 in young HDFs. Indeed, young HDFs were taken as control in all figures except this one.

<Answer>

First of all, we agree with the reviewer’s suggestion. Including the quantitative analysis for Ki67 expression, other cell proliferation assays such as BrdU staining may be helpful to demonstrate the mitogenic effect of iPSC-CENVs. However, the WST-8 assay performed in this study is a representative method for determining cell proliferation. Although it was not obvious in senescent HDFs due to their limited proliferative potential, microscopic examination of young HDFs in bright field also clearly indicated that cell proliferation was significantly induced by the treatment with iPSC-CENVs (data not shown). Moreover, as p53 and p21 play a crucial role in cell growth inhibition at upstream stages of signaling pathway, we consider that the stimulatory effect of iPSC-CENVs on cell proliferation can be verified by examining the reduced expressions of p53/p21 in senescent HDFs.

Meanwhile, as described in our manuscript, it has been reported that p53/p21 expressions in senescent cells are increased. Since the senescence of HDFs was already confirmed by observing their morphological change and elevated SA-β-Gal activity, we compared p53/p21 expressions of senescent HDFs with and without iPSC-CENVs treatment using qPCR analysis.

Minor comments: 
1. Lane 118 to 120: “This result is in line with the previous results of plasma membrane-derived nanovesicles prepared from other cell types, including several immortalized cell lines, primary cells, and stem cells”: add references.

<Answer>

In accordance with the reviewer’s comment, we added the following references after the sentence.

(Page 4, Line 121)

Additional references; [34-37]

Figure 3A: It would be more readable if the scale of the Y axis would be similar between both graphs.

Moreover, authors may compare the area under the curve.

<Answer>

In accordance with the reviewer’s comment, we created a graph showing the size distribution of both iPSC-CNEVs and iPSC-EVs at once as follows.

The main purpose of the NTA was to exhibit the size distribution of each NVs. When the concentration and size distribution were measured by NTA, CENVs and EVs derived from the same number of iPSCs were diluted in different volume of PBS. This was because there was a significant difference in the particle number of each NV sample, while the concentration of each EV had to be within the reliable detection range in the NTA. That reveals that direct comparison of the concentration value (Y-axis) for each NV in the NTA results (Figure 3A) is inappropriate. We are concerned that the readers of this article may misjudge the concentration of each NV if the size distributions are represented in a single graph. For the same reason, comparing the areas under each curve can cause this misapprehension.

For the reason, although a single graph including the size distribution of both iPSC-CENVs and iPSC-EVs is represented here, we did not change Figure 3A. We sincerely hope that the reviewer understands our consideration (please see the attachement for Figure 3A (revised)).

Reviewer 2 Report

The article IJMS-650922: “Derivation of Cell-Engineered Nanovesicles from Human Induced Pluripotent Stem Cells and Their Protective Effect on the Senescence of Dermal Fibroblasts ”, the authors studied the effects of iPSC-CENV on physiological alterations of human dermal fibroblasts (HDFs) compared to that of iPSC-Exo by natural senescence.

In this paper, the fundamental concepts and problems regarding the current state of understanding of iPSC-Exo are appropriately summarized

I would suggest that the authors include flow cytometric analysis of the cell surface markers profile from iPSCs and iPSC-CENVs to verify the pluripotent gene expression and the nanoparticle final purity.

What is the effect of the iPSC-CENV treatment on the skin aging markers, es. collagen type I and others, in senescent HDFs ?

Author Response

I would suggest that the authors include flow cytometric analysis of the cell surface markers profile from iPSCs and iPSC-CENVs to verify the pluripotent gene expression and the nanoparticle final purity.

<Answer>

In the studies that prepared CENVs by mechanically disrupting plasma membranes of various cell types, it has been demonstrated that the CENV expresses various cell surface markers, including EV-related markers such as CD63 and CD9 [1-3]. In particular, Jang et al. revealed that the treatment of the CENVs with trypsin resulted in degradation of plasma membrane proteins, indicating that the cell surface proteins were present on the surface of CENVs [4]. Furthermore, some recent studies have reported that CENVs generated from a specific type of stem cells expresses its specific markers. For example, MSC-derived NV expressed MSC-specific membrane proteins such as CD105 and CD29 [5] and mESC-derived NV expressed specific membrane proteins found on the surface of ESCs, such as ICAM-1 [6]. These results imply that iPSC-CENV also expresses human pluripotent stem cell-specific membrane proteins on its surface.

Meanwhile, the purity of NVs secreted from cells (e.g. exosomes, EVs) or generated by mechanically disrupting cell membrane (e.g. CENVs) has been evaluated by determining the particle number-to-protein ratio in many previous studies. Although it is difficult to identify other kinds of impurities, such as RNA and phospholipid, this method is a representative way to analyze the purity of membrane-derived NVs. For the reason, we assessed the purities of both iPSC-CENV and iPSC-EV sample by determining the particle number-to-protein ratio in the present study.

Kim, Y.-S.; Kim, J.-Y.; Cho, R.; Shin, D.-M.; Lee, S.W.; Oh, Y.-M. Adipose stem cell-derived nanovesicles inhibit emphysema primarily via an FGF2-dependent pathway. Exp. Mol. Med. 2017, 49, e284. Kim, H.Y.; Kumar, H.; Jo, M.-J.; Kim, J.; Yoon, J.-K.; Lee, J.-R.; Kang, M.; Choo, Y.W.; Song, S.Y.; Kwon, S.P.; Hyeon, T.; Han, I.-B.; Kim, B.-S. Therapeutic efficacy-potentiated and diseased organ-targeting nanovesicles derived from mesenchymal stem cells for spinal cord injury treatment. Nano Lett. 2018, 18, 4965-4975. Goh, W.J.; Zou, S.; Ong, W.Y.; Torta, F.; Alexandra, A.F.; Schiffelers, R.M.; Storm, G.; Wang, J.-W.; Czarny, B.; Pastorin, G. Bioinspired cell-derived nanovesicles versus exosomes as drug delivery systems: a cost-effective alternative. Sci. Rep. 2017, 7, 14322. Jang, S.C.; Kim, O.Y.; Yoon, C.M.; Choi, D.-S.; Roh, T.-Y.; Park, J.; Nilsson, J.; Lotvall, J.; Kim, Y.-K.; Gho, Y.S. Bioinspired exosome-mimetic nanovesicles for targeted delivery of chemotherapeutics to malignant tumors. ACS Nano 2013, 7, 7698-7710. Han, C.; Jeong, D.; Kim, B.; Jo, W.; Kang, H.; Cho, S.; Kim, K.H.; Park, J. Mesenchymal stem cell engineered nanovesicles for accelerated skin wound closure. ACS Biomater. Sci. Eng. 2019, 5, 1534-1543. Jo, W.; Jeong, D.; Kim, J.; Park, J. Self‐Renewal of Bone Marrow Stem Cells by Nanovesicles Engineered from Embryonic Stem Cells. Adv. Healthc. Mater. 2016, 5, 3148-3156.

In accordance with the reviewer’s comment, we added the sentences in Results section and the references as follows;

(Page 3, Line 101)

“It has been reported that CENVs generated from certain cell types contain their specific markers [32, 33].”

(Page 4, Line 121)

Additional references; [34-37]

What is the effect of the iPSC-CENV treatment on the skin aging markers, es. collagen type I and others, in senescent HDFs?

<Answer>

In the previous study, we examined the effect of iPSC-Exo on expression of collagen type I (COL1) and matrix metalloproteinase (MMP) in both photo-aged and naturally senescent HDFs. Although it was shown that iPSC-Exo restored the altered expression of collagen type I and MMP-1/3 induced by not only UVB irradiation but also replicative senescence, the extent of alterations in the expression of collagen type I and MMP-1/3 was significantly smaller in natural senescence than in photo-aging. For the reason, we investigated the effect of iPSC-CENVs on expression of p53/p21 in the present study.

In addition, the correlation between p53 and the alterations in expression of various types of collagen and MMP has been reported [7]. Struewing et al. demonstrated that the basal expression of MMP-1 was significantly decreased by down-regulation of p53 expression by transfection with specific siRNAs in endothelial cells [8]. Zhu et al. revealed that knockdown of p53 not only attenuates fibroblast senescence but also leads to a decrease in the expression of p21 and MMP-2/9, and an increase in the expression of collagen type I/III in hypoxia-treated cardiac fibroblasts or in mouse heart tissues after infraction [9]. These results indicate that p53/p21 may regulate the expression of collagen type I and MMPs at upstream levels of senescence-associated cellular signaling. Thus, we quantitatively assessed the expression of p53/p21 to demonstrated the alleviatory effect of iPSC-CENVs on the replicative senescence of dermal fibroblasts.

Coppé, J.-P.; Desprez, P.-Y.; Krtolica, A.; Campisi, J. The senescence-associated secretory phenotype: the dark side of tumor suppression. Annu. Rev. Pathol. Mech. Dis. 2010, 5, 99-118. Struewing, I.T.; Durham, S.N.; Barnett, C.D.; Mao, C.D. Enhanced endothelial cell senescence by lithium-induced matrix metalloproteinase-1 expression. J. Biol. Chem. 2009, 284, 17595-17606. Zhu, F.; Li, Y.; Zhang, J.; Piao, C.; Liu, T.; Li, H.-H.; Du, J. Senescent cardiac fibroblast is critical for cardiac fibrosis after myocardial infarction. PLoS One 2013, 8, e74535.

In accordance with the reviewer’s comment, we added the sentences in Results section and the references as follows;

(Page 9, Line 244)

“Furthermore, altered expression of various types of collagen and matrix metalloproteinase (MMP) has been found to be associated with p53 expression in senescent cells [45, 46]. Zhu et al. reported that knockdown of p53 in senescent cardiac fibroblasts of mouse heart increased the expression of collagen type I and III, but decreased the expression of MMP-2 and MMP-9 [47]. Thus, we”

Round 2

Reviewer 1 Report

I would like to thank the authors for completing certain parts of the manuscript, thus making it more readable.

However, I regret that they did not quantify by quantitative RT-PCR the expression of the pluripotency genes (Figure 2C) by indicating the Ct values ​​of iPSc and CENV. It would be informative and maybe would confirm the paper of Yoon et a.,